

# Improving hydrologic models for predictions and process understanding using Neural ODEs

Marvin Höge[1], Andreas Scheidegger[1], Marco Baity-Jesi[1], Carlo Albert[1], and Fabrizio Fenicia[1]

[1]Department of Systems Analysis, Integrated Assessment and Modelling, Eawag, Dübendorf, Switzerland

**Correspondence:** Marvin Höge (marvin.hoege@eawag.ch)

**Abstract.** Deep learning methods have frequently outperformed conceptual hydrologic models in rainfall-runoff modelling. Attempts of investigating the internals of such deep learning models are being made but traceability of model states and processes and their interrelations to model input and output is not yet fully given. Direct interpretability of mechanistic processes has always been considered as asset of conceptual models that helps to gain system understanding aside of predictability. We introduce hydrologic Neural Ordinary Differential Equation (ODE) models that perform as well as state-of-the-art deep learning methods in stream flow prediction while maintaining the ease of interpretability of conceptual hydrologic models. In Neural ODEs, internal processes that are represented in differential equations are substituted by neural networks. Therefore, Neural ODE models enable fusing deep learning with mechanistic modelling. We demonstrate the basin-specific predictive performance for several hundred catchments of the continental USA. For exemplary basins, we analyse the dynamics of states and processes learned by the model-internal neural networks. Finally, we discuss the potential of Neural ODE models in hydrology.

## 1 Introduction

### 1.1 Machine Learning in Hydrology

Deep learning models, in particular long-short-term memory (LSTM) neural networks, have outperformed traditionally used conceptual models in hydrologic modelling (Kratzert et al., 2018; Feng et al., 2020; Lees et al., 2021a). Machine learning methods provide great versatility (Shen, 2018; Shen et al., 2018; Reichstein et al., 2019) and have demonstrated unprecedented accuracy in various modelling tasks like predictions in un-gauged basins (PUB, e.g. Kratzert et al., 2019b; Prieto et al., 2019), in transfer learning to data-scarce regions (Ma et al., 2021) or flood forecasting (Frame et al., 2021; Nevo et al., 2021). Nonetheless, deep learning remains a field of progress with gaps to fill. We want to focus on three of them that are particularly relevant in hydrology.

First, machine learning models are still not as easily interpretable as traditionally-used physics-based conceptual hydrologic models are. Although high predictive accuracy is crucial to all modeling tasks, it is often not the only purpose. Especially when dealing with complex systems as it is the case in hydrology, learning about the system and understanding its internal and external interrelations is just as important to many researchers. There have been first attempts in this direction by investigating





what happens inside machine learning models (Kratzert et al., 2019a). Generally, research on explainable artificial intelligence (XAI) or "interpretable machine learning" (e.g. Samek et al., 2019; Montavon et al., 2018; Molnar et al., 2020; Molnar, 2020) has strongly advanced in recent years. Specifically, in hydrologic modelling, ties between internal model states and hydrological processes are being elicited (Lees et al., 2021b).

Therefore, it becomes more and more inaccurate to label machine learning methods as black boxes since techniques exist that shed light on the internals of machine learning methods (see also Nearing et al., 2021; Frame et al., 2021) - turning them toward so-called grey box models. Yet, internal investigation of machine learning models relies on additional methods that come with their own assumptions and caveats, and the current straight-forward interpretability of conceptual models serves as benchmark in the hydrologic community. Much environmental research is dedicated toward extrapolation in space, in time and

of boundary conditions, in order to investigate extreme events (Frame et al., 2021), climate change projections (Nearing et al., 2019) and so on. In all these fields ease of interpretability is desirable.

       Second, while the introduction of system memory as physical principle (like in LSTM models) turned out to be crucial for hydrograph prediction, other basic physical principles are not necessarily fulfilled, yet. Currently used machine learning

approaches are limited to fixed time steps that restricts their usage. For instance, while LSTM approaches work well on daily timescales, high-flow events often occur on a higher temporal resolution and can therefore not be as well resolved. For LSTM models, recent developments show adoption approaches to finer time intervals, as from daily to hourly (Gauch et al., 2021) or introduce continuous-time hidden-states within the LSTM framework in order to internally update their step-wise dynamics (Lechner and Hasani, 2020). Yet, thereby, modelling becomes more interlaced, and computational effort increases without

increasing system understanding. Despite all the progress in this field, real-world systems with their states and processes are time-continuous and from a physics perspective it remains unsatisfactory when models are restricted to certain time scales. Further, attempts to enforce fundamental principles like mass-balance were made but showed that this constraint might even worsen predictive power compared to the unconstrained LSTM variants Hoedt et al. (2021).

Third, there is often prior knowledge that cannot be included into machine-learning models. Data-driven modelling demonstrates impressive abilities in terms of mimicking and/or improving the translation from driving forces variables through the system into its output, like from precipitation to discharge in hydrology. Yet, the question remains why such models have to use only data to learn all the internals of the system from scratch. Much knowledge about hydrology has been gathered in the past so why not providing such knowledge, e.g., mechanistic structure, reliable causal interrelations and context-specific

information (Rackauckas et al., 2020), to the models directly? Of course, the risk impends that certain constitutive relations as they are used in mechanistic processes might be inexact or misleading. Nonetheless, on the one hand we can rely on many basic principles that are generally agreed upon, and on the other hand, including constitutive relations has the potential of providing additional knowledge on hydrological processes aside from data alone.





## 1.2 Conceptual Hydrologic Models

For conceptual hydrologic models, these gaps have been mostly closed over the last decades: The development of conceptual bucket-type models rests on the deductive insight that physical principles do hold in general. Basic building blocks have been elicited and modular frameworks allow to tailor models for any task at hand (Fenicia et al., 2011; Clark et al., 2015) while maintaining full interpretability of each element. Knowledge about local conditions is used to improve the models (Gnann et al., 2021), fostering both system understanding and accuracy in predictions (Kirchner, 2006; Fenicia et al., 2014) in typically data-limited modelling tasks (Fenicia et al., 2008; Li et al., 2021).

Yet, there remains a dichotomy between bottom-up and top-down approaches in hydrology (Savenije, 2009; Gharari et al., 2021). In the former, process knowledge that was acquired at smaller scales is generalized to catchment scale while in the latter prediction and interpretation of the hydrologic system is based on the overall catchment response (Sivapalan et al., 2003). The bottom-up approach yields physically-based and distributed models (Abbott et al., 1986; Loritz et al., 2018) and, over recent years, different methods have been investigated to learn constitutive equations directly from data (Gharari et al., 2021). Top-down models have been widely explored using different modelling approaches which include the present range of conceptual model structures (see Knoben et al., 2020), aided by flexible frameworks such as Superflex (Fenicia et al., 2011) or FUSE (Clark et al., 2008), or transfer function models (Young, 2003). Both approaches seek to obtain parsimonious models that shall be as simple as possible for the sake of interpretability and complex enough to achieve high predictive accuracy (Höge et al., 2018; Gharari et al., 2021). Often enough, only a few model states and processes (see, e.g. Patil and Stieglitz, 2014) are sufficient as effective theory to describe an entire hydrologic system (Kirchner, 2009; Fenicia et al., 2016). However, the plethora of hydrologic models itself points at the fact that no single model or framework exists that is always applicable.

Recently, attempts have been made to develop models that fuse both model types in order to alleviate the shortcomings of each type (e.g., Zhao et al., 2019; Bennett and Nijssen, 2021). For example, Jiang et al. (2020) used convolutional neural networks (CNN) to predict discharge time series taking meteorological input time series and the output from a conceptual hydrologic model as inputs. There, the hydrologic model output acts as physical guidance that is used aside of driving input forces like precipitation by a CNN to achieve better discharge predictions. In their workflow, the application of the neural network serves as a postprocessing to the hydrologic model simulation. Therefore, we refer to this approach as external hybrid modelling.

## 1.3 Scientific Machine Learning and Neural ODEs

Yet, none of the pure or novel hybrid machine learning approaches has addressed all the above gaps regarding interpretability, physics and knowledge at once. Here, we introduce a different hybrid modelling approach that is able to close them simultaneously and that also has the potential to help dissolving the dichotomy in hydrology. We employ Neural Ordinary Differential





Equation (ODE) models (Chen et al., 2018; Rackauckas et al., 2020), i.e., models based on differential equations with terms that are substituted by neural networks partially or entirely. Neural ODEs fuse mechanistic physics with machine learning and their appeal is twofold: First, differential equations as mathematically elegant representations of scientific interrelations have been well investigated and widely used. Neural ODEs extend this framework. Second, it is much easier for a neural network to not learn the behaviour of the observable directly but to encode the mechanism behind that determines the observed behaviour (Rackauckas et al., 2020). In other words, the derivatives often have simpler functional relationships than their solution. Comparably simple mechanistic interrelations sometimes lead to very complex observable outcomes like, e.g., chaos.

On a broader scope, this is the field of scientific machine learning introduced by Rackauckas et al. (2020) that seeks to bring together both the knowledge contained in data (bottom-up) and knowledge from expertise (top-down), and leverage both for greater knowledge gain, higher predictive power and increased system understanding. The rationale behind scientific machine learning is that reliable inter- and extrapolation in science has always overwhelmingly been due to mechanistic laws that impose physical structure to the problem at hand. With pure data-driven approaches this structure has to be learned entirely from data. Here, the inclusion of mechanistic principles might help to fill knowledge gaps, especially in data-limited contexts, and novel differentiable programming tools foster its application in scientific computing (Innes et al., 2019). In scientific machine learning it is possible (and desired) to include physical structure and processes that are known mechanistically as hard-coded features and leave what is not known or only known vaguely to the data-driven method.

Deep learning methods in hydrology have proven their ability to process integrated site-specific information to improve discharge prediction tremendously (Kratzert et al., 2019c). This has not been possible with conceptual models. Nonetheless, there might be catchments with unique features or site-specific conditions that are invisible to machine learning methods due to only using averaged attributes or due to the fact that these features are exceptions and distinctively different from any other basin. Further, it might be impossible to provide respective information (like highly resolved spatially explicit features) to a machine learning method since it becomes computationally infeasible. Pure machine learning approaches are not meant to be modified by adding specifics via hard-coding additional formulas into the model. Contrarily, scientific machine learning provides an interactive framework where knowledge can be included explicitly, allowing us to "put humans into the loop" (e.g. Holzinger, 2016) if desired, and not to leave this resource of knowledge aside. This pertains to, e.g., identifying plausible processes based on mechanistic understanding, or to providing context information: seasonal features, specific topography, geology (e.g. karst), and so on. We introduce scientific machine learning for hydrology by leveraging a physics-based conceptual hydrologic model with one or several neural networks, substituting mechanisms in the underlying mass-balance ODEs.

The remainder of this article is structured as follows: In Section 2, we introduce our model and the used data as well as the chosen training and evaluation procedure. In Section 3, we rate predictive accuracy of our models on a few common hydrologic metrics. There, we present our internal hybrid approach in direct conjunction to state-of-the-art results from Jiang et al. (2020).



In Section 4 we analyse model internal states and processes dynamics of our Neural ODE models. We discuss the results and their implication in Section 5. Finally, we close with a conclusion and outlook in Section 6.

## 2 Methods

### 2.1 Models

As baseline conceptual framework, we work with a typical hydrologic bucket-type model. We employ the structure of the simple rainfall-runoff model EXP-Hydro (Patil and Stieglitz, 2014). The model comprises only two state variables as buckets: snow storage $S_{snow}$ and so-called catchment water storage $S_{water}$; and five mechanistic processes: precipitation of rain $P_{rain}$ and snow $P_{snow}$, melting $M$, evapotranspiration $\widetilde{ET}$ and discharge $Q$. In general terms, the coupled ODE model structure writes as:

$$\frac{\mathrm{d}S_{snow}(t)}{\mathrm{d}t} = P_{snow}(\mathbf{I}(t); \boldsymbol{\Theta}) - M(\mathbf{I}(t); \boldsymbol{\Theta}) \tag{1}$$

$$\frac{\mathrm{d}S_{water}(t)}{\mathrm{d}t} = P_{rain}(\mathbf{I}(t); \boldsymbol{\Theta}) + M(\mathbf{I}(t); \boldsymbol{\Theta}) - L_{day}(t) \cdot \widetilde{ET}(\mathbf{I}(t); \boldsymbol{\Theta}) - Q(\mathbf{I}(t); \boldsymbol{\Theta}) \tag{2}$$

with time $t$, length of day $L_{day}(t)$ and model parameters $\boldsymbol{\Theta}$. Model inputs and internal states are defined as $\mathbf{I}(t) = (T(t), P(t), S_{snow}(t), S_{water}(t))^{\mathrm{T}}$, with temperature $T(t)$ and precipitation $P(t)$ as driving forces and model states $S_{snow}(t)$ and $S_{water}(t)$. Depending on the process, not every element in the generally formulated $\mathbf{I}(t)$ might be used. Note that the actual estimated evapotranspirative flux ET is $\widetilde{ET}$ multiplied by $L_{day}(t)$. The conceptual model structure is shown schematically in Figure 1 (a).

EXP-Hydro as originally developed by Patil and Stieglitz (2014) and re-implemented by Jiang et al. (2020) is discretized for daily time steps. Opposed thereto, we use a solver with adaptive time stepping (see Rackauckas and Nie, 2017). Since input data series are only available with fixed observation times, we apply monotonic interpolation using Steffen's method (Steffen, 1990). To foster comparability, we use EXP-Hydro as implemented by Jiang et al. (2020) as starting point but transferred it to the programming language Julia (Bezanson et al., 2017). All original equations of the five mechanistic processes with process-specific driving forces, model states and model parameters can be found in Appendix A1. Note that, there, the precipitation terms in Equations 1 and 2 do not have any dependence on model states while discharge only depends on the model state $S_{water}(t)$.

We refer to our implementation of EXP-Hydro as model M0. In total, we set up three different models with numbers in the model name indicating the percentage of neural network fraction within the model. Our models M50 and M100 have terms in Equations (1) and (2) substituted by feed-forward neural networks. To build M50, we replaced the mechanistic formulas of evapotranspiration and discharge by two small neural networks, $\mathrm{NN}_{ET}^{50}$ and $\mathrm{NN}_{Q}^{50}$, respectively. As indicated in Figure 1 (b), both NNs have two hidden layers with 16 nodes each, one output node and input nodes for all driving forces variables and





model states that are considered relevant. Compared to the plain mechanistic process (see Equation A3), also $S_{snow}$ is an additional input to the $\widetilde{ET}^{50}$, accounting for any interference of snow cover with evapotranspirative fluxes. Regarding discharge $Q^{50}$, precipitation (without specification about whether as rain or as snow) serves as an additional input, potentially allowing

the network to emulate processes like direct surface run-off.

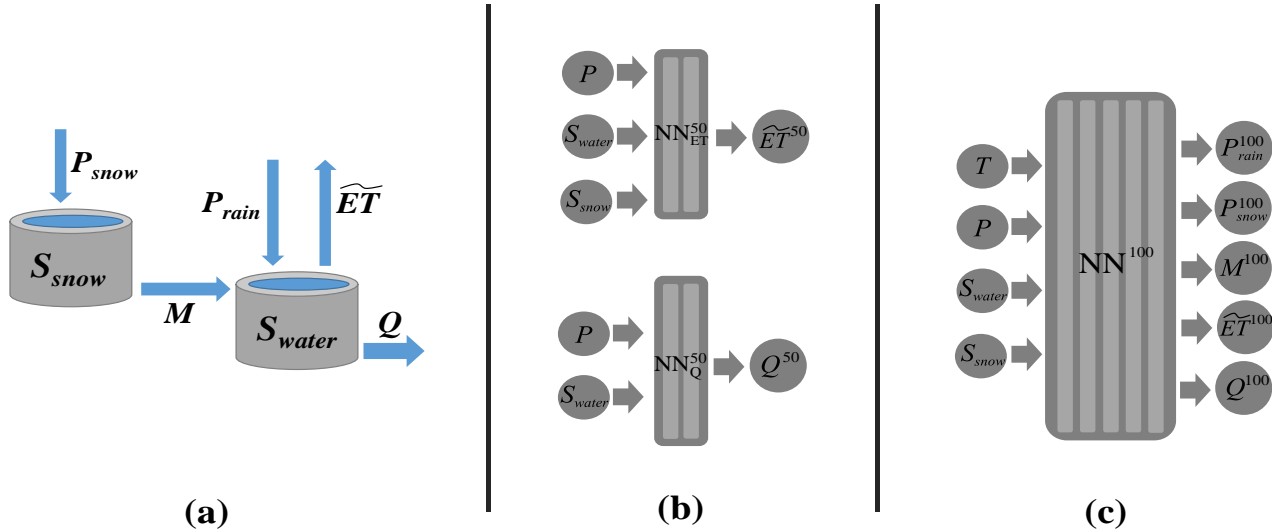

**Figure 1.** Scheme of (a) the conceptual model (M0), (b) the neural networks in M50, and (c) the single neural network in M100: (a) The model structure with 2 model states and 5 processes; (b) The two small neural networks in M50 that substitute evapotranspiration ($\text{NN}_{ET}^{50}$) and discharge ($\text{NN}_Q^{50}$) have only one output node and each has an additional input variable compared to the basic mechanistic process. (c) The large neural network ($\text{NN}^{100}$) has five output nodes - one for each substituted process - and all driving forces and model states as input.

As shown in Figure 1 (c), M100 contains only one single neural network $\text{NN}^{100}$ with five output nodes substituting all mechanistic processes in the model. M100 has both external input variables temperature and precipitation as well as the internal states of snow and water storage as inputs. The neural network has five hidden layers, each with 32 nodes, and the five
model processes in Equations (1) and (2) are replaced by output nodes $P_{snow}^{100}(t)$, $P_{rain}^{100}(t)$, $M^{100}(t)$, $ET^{100}(t)$ and $Q^{100}(t)$, respectively. A more detailed rationale for developing models M50 and M100 from M0 is available in Section A2.

## 2.2 Data

We use the data provided in the CAMELS-US dataset (Addor et al., 2017) that contains catchment-specific uniformly organized data for 671 catchments. The dynamic time series in this dataset have a daily resolution. Besides the discharge time





series, they cover also the three input forcing variables to the model: day length, temperature and precipitation. Specifically, the forcings are based on the Daymet data set, that has the spatial highest resolution (1km x 1km) compared to the available alternatives (Newman et al., 2015, and references therein). Daymet was also used by Jiang et al. (2020) and it showed to give the best results among the alternative input data sources in other modelling attempts (Newman et al., 2015; Kratzert et al., 2021).

In our model evaluation, we also use lumped snow water equivalent (SWE) time series data for each basin. Aside of the catchment-integrated time series such as those for temperature or precipitation, the CAMELS dataset contains dynamic data provided for different elevation bands in each basin, including SWE time series. Each elevation band is assigned a respective area as fraction of the full catchment area. Using this information, we integrate the SWE data as an area weighted average in order to obtain lumped SWE data for each catchment. Note that SWE is not used as model input in calibration. The observed

SWE are solely used for comparison with the dynamics of the snow storage $S_{snow}$ of the models.

From the 671 available catchments, we use the same 569 as in Jiang et al. (2020). Likewise, the calibration/training period is set to 1.10.1980 - 30.9.2000 and the validation/test period to 1.10.2000 - 30.9.2010, comprising of 20 and 10 hydrologic years, respectively. Model evaluation is based on the validation period only.

**2.3  Procedure and Model Rating**

Our models are calibrated to each catchment specifically and validated on the same catchment. The procedure is structured as follows, with steps 2 and 3 only applying to Neural ODE models:

1. Conceptual hydrologic model training: M0 is calibrated with the training data using only the Nash-Sutcliffe efficiency (see Equation (3) below) as objective function.

2. Neural network pre-training: Each internal process of the calibrated M0 is simulated individually using the required driving variables or simulated model states over the training period. Then, the neural network(s) that shall substitute the respective processes are trained on these simulated process data series with sum of squared errors as objective function.

3. Neural ODE model training: the pre-trained neural networks are inserted into the conceptual model framework and the entire Neural ODE model is trained on the calibration data.

Over the different steps, we enable knowledge transfer between the models: results from the trained conceptual hydrologic model are used as example for the neural network(s) to learn general relations between input variables and output quantities. These relations are then improved and refined in the Neural ODE training step. After successful training, we conduct a twofold evaluation of the models with validation data from test period between 1.10.2000 and 30.9.2010:

1. We benchmark the models by three metrics commonly used in hydrology (cf. Jiang et al., 2020) and compare them to
state-of-the-art model approaches (see Section 3).





2. We analyse internal model states and processes between the conceptual (M0) and the Neural ODE (M50, M100) models (see Section 4).

First, for benchmarking, the following metrics are used: The Nash-Sutcliffe efficiency (NSE), as defined in Equation (3) with $\alpha = 2$, compares the used model to simply using the average of observed discharges for predictions. With NSE $< 0$, the
model is worse than just using the model average, while the maximum value of 1 indicates perfect fit. Values above 0.55 are considered to represent "some model skill" (Newman et al., 2015). Generally, there is no fixed scheme to interpret NSE values but rules of thumb are available (see Moriasi et al., 2007; Schaefli and Gupta, 2007). Following Legates and McCabe Jr (1999), NSE ($\alpha = 2$) is only a special case of the so-called coefficient of efficiency over $N$ corresponding observed $Q_{obs}$ and simulated $Q_{sim}$ discharge values:

$$CoE_\alpha = 1 - \frac{\sum_{i=1}^{N} |Q_{obs,i} - Q_{sim,i}|^\alpha}{\sum_{i=1}^{N} |Q_{obs,i} - \overline{Q}_{obs}|^\alpha} \qquad (3)$$

Another special case with $\alpha = 1$ is referred to as modified coefficient of efficiency (Legates and McCabe Jr, 1999) or, briefly, as mNSE (Jiang et al., 2020). The values of mNSE ($CoE_1$) can be interpreted similarly to NSE ($CoE_2$). The mNSE, however, gives less weight to extreme fluctuations than the NSE, which typically relate to peak flow. Hence, mNSE is better suited to rate low and base flow. Peak flow is rated specifically by the percent bias in flow duration curve high-segment volume (Yilmaz
et al., 2008, FHV):

$$FHV = 100 \cdot \frac{\sum_{h=1}^{H} (Q_{sim:high,h} - Q_{obs:high,h})}{\sum_{h=1}^{H} Q_{obs:high,h}} \qquad (4)$$

,

where $Q_{obs:high}$ and $Q_{sim:high}$ refer to the sorted observed and simulated discharges in descending order, respectively. $H$ defines the number of highest values according to a chosen exceedance probability. Here, for comparability reasons, we use
the exceedance probability 0.01 like in Jiang et al. (2020). This means that FHV is based on the highest percent of discharges, opposed to the typical chosen exceedance probability of 0.02 (Yilmaz et al., 2008). The optimal value of FHV is 0. For comparability and since FHV values can become negative, we use only the absolute values like in Jiang et al. (2020) (where FHV was renamed to absolute peak flow bias PFAB).

Second, the evaluation of internal model states and processes is conducted in direct comparison between the conceptual model M0 and the Neural ODE models M50 and M100: The dynamics of snow and water storages is inspected alongside the model-specific estimated streamflow. Further, the internal processes for discharge, evapotranspiration and melting are isolated and explored over plausible ranges of input variables and model states, e.g. discharge as a function of water storage. Additional input variables to the neural networks in M50 and M100 that shall not be explored are kept fixed with catchment-specific values
(like mean temperature) as specified in Section 4.





## 3 Benchmarking Neural ODE Models

Figure 2 shows the distributions of the three evaluation metrics per evaluated model over all 569 considered catchments. NSE, FHV and mNSE are displayed in one row per model, i.e. the two newly developed Neural ODE models (M50 and M100), the conceptual model M0, and two state-of-the-art models. The shown performance values for both the hybrid CNN and LSTM

model are the original values from Jiang et al. (2020).

**Figure 2.** Histograms of NSE (red; optimal value: 1), FHV (green; optimal value: 0) and mNSE (blue; optimal value: 1) for the developed Neural ODE models M100 and M50, the plain conceptual baseline model M0 and state-of-the-art LSTM and external hybrid CNN models (bottom, cf. Jiang et al., 2020).

For both FHV and mNSE, the M100 scores better in both mean and median than all other models. The distributions over all catchments show clear shifts towards the optimal scores 0 for FHV and 1 for mNSE, respectively. Considering NSE, which





is also the calibration metric, M100 outperforms all other models except for the hybrid CNN approach. Yet, both mean and median NSE between the two models do only deviate by a small margin. Looking at the histograms, it can be seen that the

hybrid CNN model shows an accumulation of scores slightly above the median for NSE and mNSE and slightly below the median for FHV. Contrarily, the M100 achieves significantly higher scores for NSE and mNSE and lower peak flow errors. At the tails of the histograms, especially M100 managed to reduce the number of bad results (NSE and mNSE below 0 and FHV around 100 and above).

Considering M50 and M0, the Neural ODE model M50 achieves a significant improvement in all metrics over the plain conceptual model: NSE mean and median improve by about 0.15 and 0.23, respectively; mNSE increases in both statistical moments by more than 0.1, while FHV drops by about $25\%$. This shows that the conceptual model significantly benefits already from substituting only two processes ($ET$ and $Q$) by more flexible methods.

It can easily be seen that all models except for model M0 and LSTM achieve performances in a similar range with similar means and medians over all metrics although the distributions show noticeable differences. While M0 shows better FHV scores with the whole distribution tending toward lower values, the LSTM is significantly better regarding NSE and mNSE. Yet, all distributions for both models deviate clearly from the other models, showing significantly more bad values that are low (around 0.0) for NSE and mNSE and high for FHV. This is further discussed in Section 5.1 with a special focus on LSTM models.


## 4 Internal States and Processes of Neural ODE models

As with conceptual hydrologic models, the temporal dynamics of processes and states can directly be inspected and analysed in the Neural ODE approach. We chose two exemplary basins for demonstration purposes: Fish River near Fort Kent, Maine (ID: 1013500) and Spearfish creek, South Dakota (ID: 6431500). Figure 3 shows the time series of discharge, snow storage

and water storage states from the plain conceptual (M0) and the Neural ODE models (M50 and M100) for both basins. For discharge and snow storage, observations are available and are displayed, with the latter being the lumped snow water equivalent (SWE) data. Note, that SWE was not used in the calibration.

The two basins cover different magnitudes for all depicted variables. For the basin 1013500, model predictions of the three

models are very similar. Discharge predictions of all models match observations very well which is also indicated by overall good metrics in Table 1. The agreement between models is weaker for snow storage although the general pattern is similar and approximately matches observations. For basin 6431500, model predictions deviate more strongly and show a larger discrepancy to data. As supported by rather bad performance metrics, model M0 underestimates baseflow in large parts and misses both timing and flashiness of peaks.






**Table 1.** Streamflow prediction performance based on NSE (optimum: 1), FHV (optimum: 0) and mNSE (optimum: 1) of the conceptual model (M0) and both Neural ODE models (M50 and M100) for basins 1013500 and 6431500.

| Model | Basin 1013500 | | | Basin 6431500 | | |
|---|---|---|---|---|---|---|
| | NSE | FHV | mNSE | NSE | FHV | mNSE |
| M0 | 0.85 | 8.38 | 0.66 | 0.005 | 34.73 | -0.28 |
| M50 | 0.89 | **4.94** | 0.7 | 0.33 | 30.47 | 0.18 |
| M100 | **0.91** | 5.41 | **0.73** | **0.54** | **9.14** | **0.19** |

In neither basin, the Neural ODE models do alter the snow storage component much from the plain conceptual model although there are small differences in specific years. Overall, the models do catch the temporal pattern of snow accumulation but there are discrepancies in the magnitude. The models for basin 1013500 show acceptable estimates while for basin 6431500 they tend to underestimate SWE systematically. At the end of each snow season, the models predict snow to disappears much

earlier compared to the observed values for most years. This issue is further discussed in Section 5.2. Regarding water storage, there is no data for a direct comparison available. For basin 1013500, all models strongly agree on the dynamics and magnitude of the model state. This is different with model estimates for the second basin where the two Neural ODEs are similar with only small deviations but both differ significantly from the conceptual model estimate. Apart from variations in the annual cycles, M50 and M100 show much smaller variance while M0 indicates a general magnitude shift to higher water storage in the last

third of the testing period. Together with the significantly better scores in Table 1 of both Neural ODE models, this indicates that M0 might not be a suitable choice as model for this particular basin.





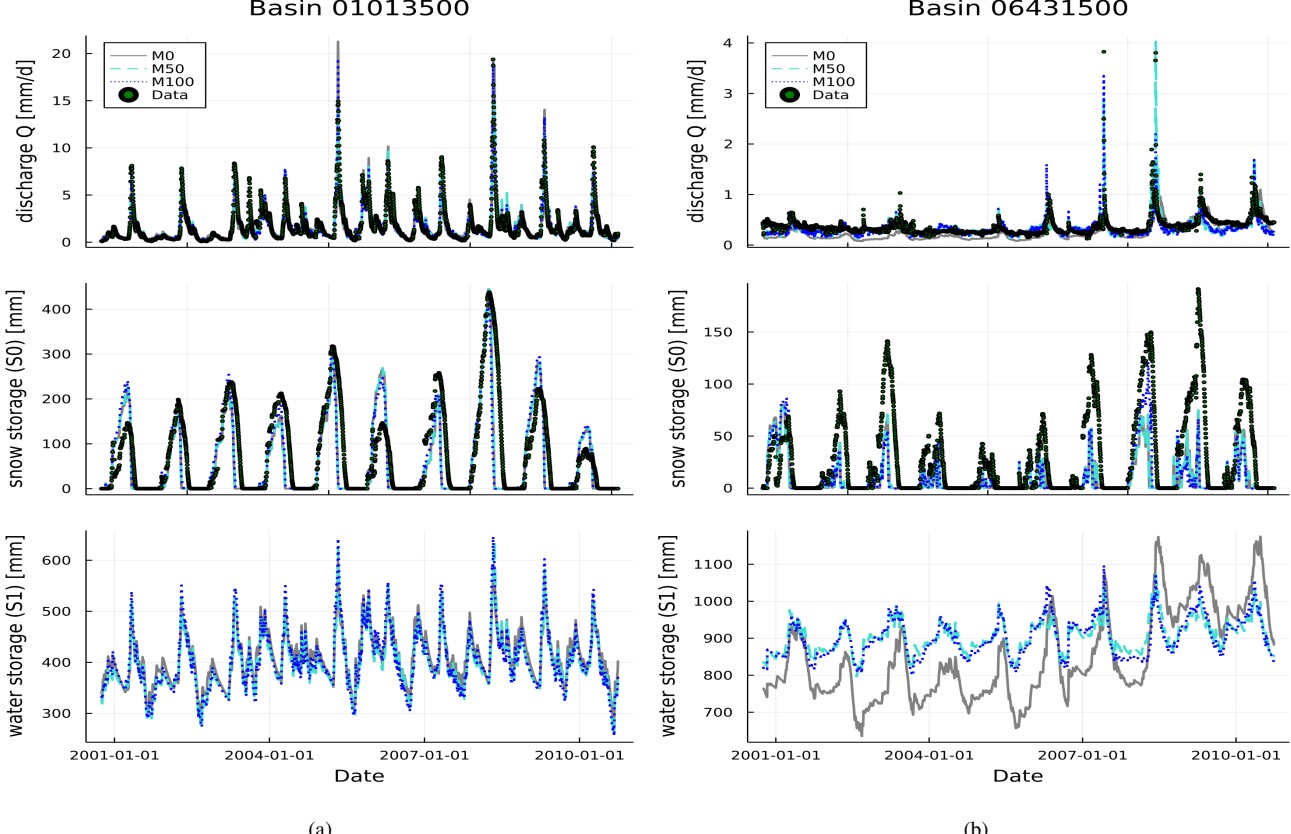

**Figure 3.** Time series of data and model predictions from models M0, M50 and M100 for discharge (top), snow storage (center) and water storage (bottom; no data) for the test period in basin 1013500 (a) and in basin 6431500 (b).

Aside of direct inspection, we analyse internal processes over plausible ranges of input variables investigating discharge, evapotranspiration and melting. Figure 4 shows the relations between water storage $S1$ and discharge $Q$ for models M0 (hard-
coded, Equation A5), M50 (learned by $NN_Q^{50}$) and M100 (learned by $NN^{100}$) for both basins.

All three discharge to water storage relations are very similar for small to medium water storage values. Beyond, strong differences evolve: In basin 1013500, the hard-coded relation in M0 shows a strong increase for large values of $S1$ reaching values much higher than the maximum discharge that was observed over both the training and the testing period. At the same
time, M50 shows a linear trend for large water storages underestimating the maximum observed discharge slightly. For basin 6431500 it is the opposite, M0 underestimates discharge and model M50 shows a strong tendency to overshoot. In both basins, M100 shows the most plausible relation.





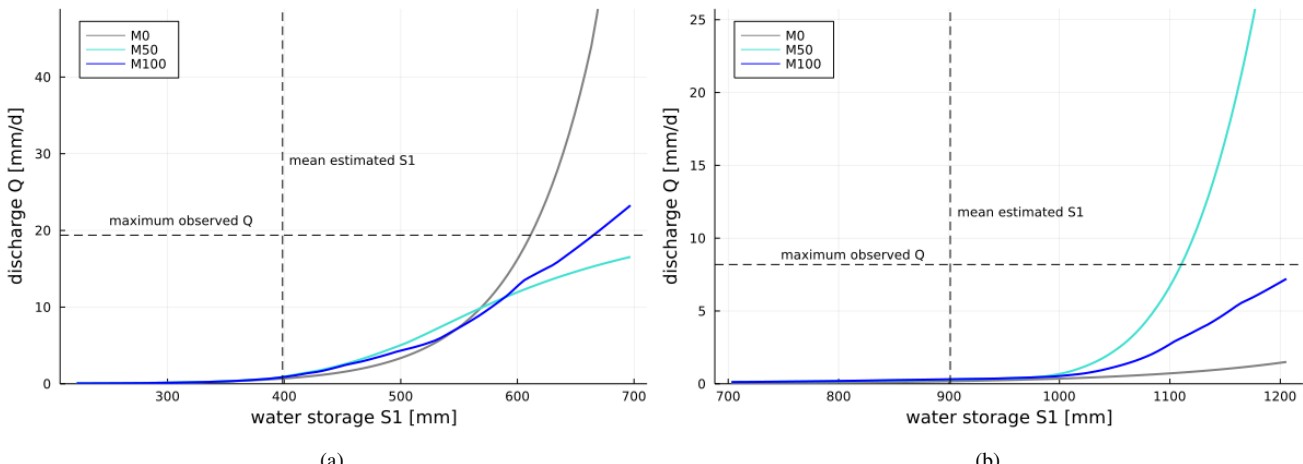

(a)            (b)

**Figure 4.** Relation between water storage and discharge in basin 1013500 (a) and basin 6431500 (b) for models M0 (hard-coded relation), M50 and M100 (learned by neural network, with additional neural network inputs snow storage fixed at 0 for both M50 and M100, and both precipitation and temperature fixed at basin averages over the training period).

The Neural ODE approach allows to directly analyse the impact of additionally assigned variables to specific processes.

Both neural networks $NN_Q^{50}$ and $NN^{100}$ in M50 and M100, respectively, also use precipitation as input. Figure 5 depicts the relations of discharge to water storage and precipitation for the three models in each basin. Note that the magnitudes of discharge vary between basins and that discharge in the conceptual model M0 depends only on water storage. For model M0, the very high discharge predictions in basin 1013500 and the very low ones in basin 6431500 are clearly shown in Figure 5 (a) and (b), respectively.


For basin 1013500, models M50 and M100 show an overall similar pattern in Figure 5 (c) and (e), respectively, with M100 reaching higher magnitudes. Both models locate the highest discharge in the same region of high water storage and medium to low precipitation. For very small precipitation at high water storage especially M100 indicates a slight decline in discharge. Interestingly, neither model shows an increase of discharge with stronger precipitation. The decline in discharge could be related

to the lower frequency of strong rain events in the basin and the resulting detriment of the neural networks to learn another relation. Hence, it might be subject to higher uncertainty in this variable range. This is further discussed in Section 5.2.

The expected trend of increasing discharge for increasing rain is clearly visible for both model in basin 6431500 (Figure 5 (d) and (f)). Notably, a peak in discharge for high water storage and small rain rates is visible similarly to the other basin.

Investigations about whether this could be an indication of a general non-linearity required further discussion (see Section 5.2).





**Figure 5.** Dependence of discharge on precipitation (rain) and water storage for Neural ODE models M0, M50 and M100 in basin 1013500 (left; (a),(c),(e), respectively) and in basin 6431500 (right; (b),(d),(f), respectively). For $NN^{100}$ the additional neural network input snow storage was fixed at 0 in both basins and temperature was fixed at the average temperatures over the training period ($7.7^{\circ}C$ for basin 1013500 and $10.52^{\circ}C$ for basin 6431500).





Figure 6 depicts the models' dependences of the evapotranspiration terms (without $L_{day}$) on temperature and water storage. Note that the magnitude ranges of evaportranspiration indicated by colors are the same between the basins. The hard-coded relation according to Hamon's formula in the conceptual model M0 shows the most regular behaviour in Figure 6 (a) and (b) for both catchments: for temperatures below $0°C$ there is very small to no evapotranspiration. Overall, increasing temperatures or water storages are associated to increasing evapotranspiration although the general magnitude is smaller for basin 6431500.

For basin 1013500, M0 shows significantly higher ET estimates over a large range of temperature-water storage combinations compared to the other two models. M50 reaches maximal ET only in the region of medium to high water storage and very high temperatures (extreme to unrealistic for the considered basin) as shown in 6 (c). The general trend of higher ET for higher temperature is also learned by the $NN_{ET}^{50}$, but the pattern is not as regular as in M0. In particular, for water storage values at the extremes, a decrease of evapotranspiration is assumed. This indicates that either the water storage-ET relation is not as proportional in these regions as assumed by Hamon's formula, or there is a lack of datapoints covering these ranges making it challenging for $NN_{ET}^{50}$ to elicit the underlying relation . Nonetheless, the elicited relation appears plausible in particular for small water storage.

In contrast to M50, M100 shows a much more regular dependence of ET on temperature and water storage as shown in Figure 6 (e). It shows the same regular increase of ET with temperature as in M0 but a smaller dependence on water storage. Yet, the magnitude of ET estimated by the neural network in M100 is generally significantly smaller. Hence, according to M100, evapotranspiration plays a generally smaller role in the water balance.

In basin 6431500, both models show a much more similar pattern for the maxima of evaportanspiration (6 (d) and (f)) although M100 indicates a much stronger increase in magnitude for rising temperatures and water storage. In contrast to Hamon's formula in M0, the neural networks do not allocate strong ET rates to small to medium water storages even for higher temperatures, but both models depict higher rates even for lower temperatures if water storage is high.





**Figure 6.** Dependence of evapotranspiration on temperature and water storage for Neural ODE models M0, M50 and M100 in basin 1013500 (left; (a),(c),(e), respectively) and in basin 6431500 (right; (b),(d),(f), respectively). For $NN_{ET}^{50}$ and $NN^{100}$ the additional neural network input snow storage was fixed at 0 in both basins, and precipitation for $NN^{100}$ was fixed at the average over the training period (3 mm/d for basin 1013500 and 1.9 mm/d for basin 6431500).



The effect of snow storage and temperature on melting rates is displayed in Figure 7. M0 and M50 employ the same hard-coded melting formula (see Equation A4) while in M100, the relation is learned by $NN^{100}$. Note that magnitudes of melting rates and snow storage range much higher in basin 1013500 than in basin 6431500. Despite some differences, there are also general trends over all models and both basins: Plausibly, no relation show snow melt for temperature below $0°C$ - this is

determined for M50 and M0 but also not altered in M100. For larger temperatures, melting rates constantly increase. The only exception are very small snow storage values where no to only slowly growing melting occurs in the models.

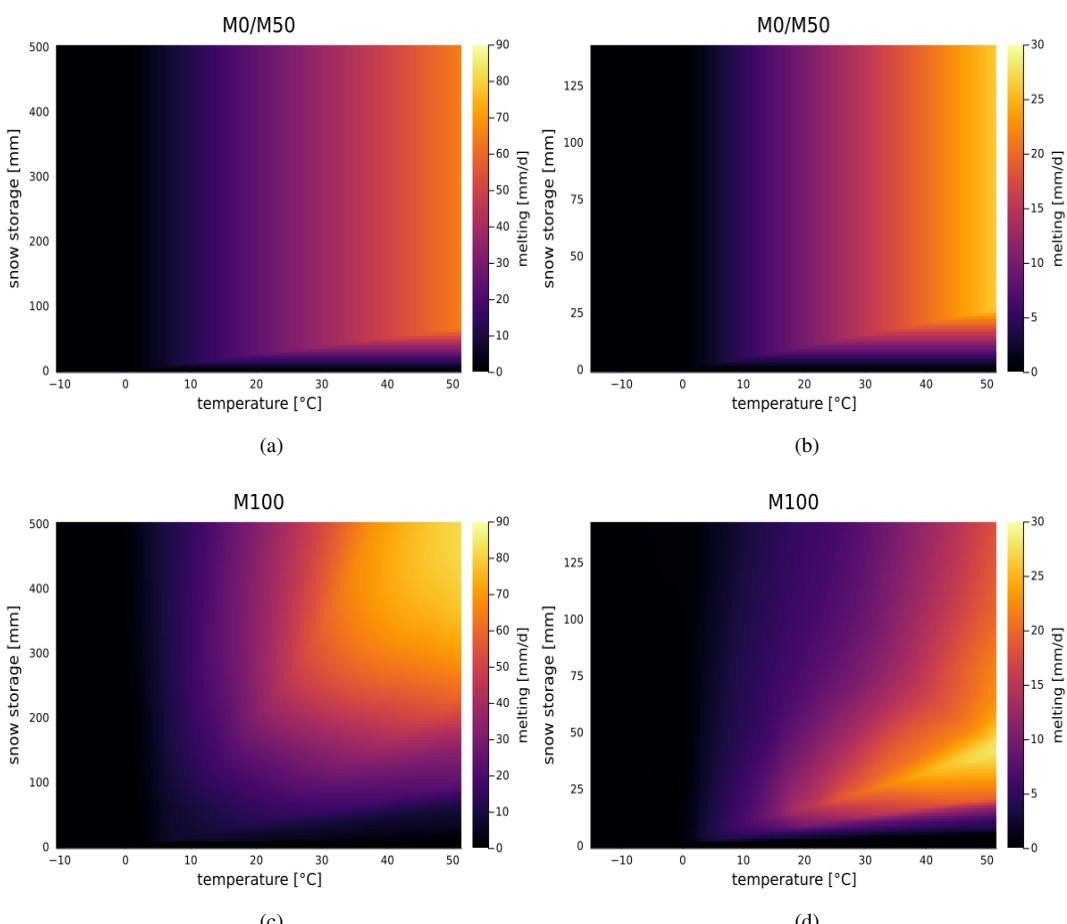

**Figure 7.** Dependence of melting rate on temperature and snow storage for Neural ODE models M0/M50 and M100 in basin 1013500 (left; (a),(c), respectively) and in basin 6431500 (right; (b),(d), respectively). For $NN^{100}$ the additional neural network inputs were fixed at basin averages of water storage and precipitation (3 mm/d for basin 1013500 and 1.9 mm/d for basin 6431500) over the training period.

For M100 differences between the basins and from the hard-coded melting linear relationship in M0/M50 are clearly observable: For basin 1013500 (Figure 7 (c)) M100 shows the smoothest increase in the direction of higher temperatures and

higher snow storage and also reaches significantly higher magnitudes. Yet, with small but growing snow storage for higher





temperatures, M0/M50 shows a stronger increase over a smaller range. This increase is similar for M100 in basin 6431500 (Figure 7 (d)) although for higher snow storage there is a decline in melting rate over the entire range for temperatures above $15°C$. Being further discussed in Section 5.2, this could again be due to a lack of training points in a catchment of warmer climate.


Of course, the highest temperatures covered in the above analysis are unrealistic to be associated to snow cover. Elevation information that would make it possible to consider snow cover in high altitudes while having already warm temperatures in lower parts of the catchment is neglected. Nonetheless, we demonstrate that a physical extrapolation and analysis of individual processes is possible with the Neural ODE approach just as it is traditionally done with conceptual models.


## 5   Discussion

### 5.1   Benchmarking

All four machine-learning based hydrologic models show a significant improvement over the plain conceptual hydrologic model M0. Results indicate that more information from training data can be leveraged by partial or pure data-driven models

and significantly higher rating scores are achieved. Arguably, the EXP-Hydro is a very simplistic bucket model and more sophisticated conceptual hydrologic models exist that achieve higher scores (see SAC-SMA in Appendix A3). Yet, more complex conceptual hydrologic models also require more tailored model features and higher parametrizations that again entail more assumptions and fine-tuning.

Note that the displayed results for LSTM are the original values from Jiang et al. (2020). There, they were obtained by catchment-specific calibration and validation. However, over recent years, LSTM models have achieved much higher scores when being calibrated to many catchments simultaneously while including also static catchment attributes (like topography, climatic indices, etc.) as additional inputs to the model (Kratzert et al., 2019c; Feng et al., 2020). LSTM models have demonstrated their ability to transfer learned relations between input variables, attributes and streamflow to unseen catchments, often

yielding highly accurate predictions. This application case of large-sample hydrology is however different from the application scenario here.

Despite their success, machine-learning models in hydrology like LSTMs are known for often underestimating high flow events (Kratzert et al., 2018). They often miss sharp peaks as they regularly occur in hydrographs. Conceptual models with

their hard-coded peak flow relations are typically very good at this task. Both Neural ODE models - and especially M100 - show a significant improvement in this respect based on learned relations that do not have to use a threshold to distinguish between base and peak flow. FHV scores of M50 are similar (yet still higher) to the hybrid CNN model that already showed an improvement in peak flow prediction taking conceptual model predictions as additional model input (see Jiang et al., 2020).





M100 achieves an even higher level of performance with a median of about 13 and a mean of about 16. The improved base
flow prediction performance is likewise indicated by highest mNSE scores (median 0.54 and mean 0.51). Overall, we summarize that Neural ODE models perform similarly well or better than alternative state-of-the art partial or pure machine learning
models.

## 5.2 Internal States and Processes

The overall better performance of Neural ODE models compared to plain conceptual models is associated to significant differences in the model internal dynamics and process relations. Results demonstrate that the pre-training of neural networks in
order to mimic hard-coded processes before the full Neural ODE training does not prevent the neural networks from learning
new and vastly different relations. With Neural ODEs being built on the same conceptual model structure, individual states and
processes can easily be analysed and compared between different models or they can be investigated over specific ranges of
input variables and models states.

In the variable ranges where much data was available the Neural ODE models elicited plausible relations for the investigated
processes. Yet, the analyses indicated that a lack of training data in the extreme ranges of the process dependent variables
increases uncertainty and might indicate counter-intuitive relations. 20 years of training data for a single catchment typically
do not provide enough information to certainly extrapolate towards these limits. Although general process trends often appeared to be plausible, there are cases (e.g. a decrease of melting rate for growing snow storage) that are hardly explainable
and presumably require more data to be refined. Having similar relations despite using more data could then help investigating
counter-intuitive process dynamics. Further, the lumped model structure with only a few processes forces parameters to falsely
take up information from data although it should inform another (not included) process. This might additionally exacerbate
the elicitation of clear relations. As remedy, a more detailed conceptual structure might improve the encoding of underlying
functional relations.

When looking at discharge and snow storage dynamics for both of which data are available, further potentials and limitations of the approach are observable: The Neural ODE models show much higher accuracy than the plain conceptual model
in stream flow prediction. For discharge in the small and medium range of water storage, Neural ODE models do not alter
the functional dependence to water storage from the conceptual model much. However, they deviate significantly for the high
regime requiring no separation between base and peak flow. Further, they allow modellers to investigate the direct impact of
other variables like precipitation on discharge. These dependencies learnt by the neural networks might help developing more
sophisticated discharge relations.


Despite the close agreement between M0/M50 and M100 regarding their predictions of snow in both considered basins, all
models depict limitations of the lumped snow storage approach: Melting of snow is often predicted earlier than shown for the





catchment by data (see Figure 3). In higher altitudes, snow stays much longer and new precipitation might also fill the snow storage there even if in lower altitudes spring and summer might already have started. Using only lumped driving forces like average temperature as input variable to the model prohibits the models to account for these effects and leads to potentially inaccurate estimates. Since the CAMELS dataset provides elevation-bands data for snow water equivalent, we assume that including elevation-resolved snow storage units in the Neural ODE models might improve this significantly.

## 6  Conclusions and Outlook

Hydrologic Neural ODE models fuse the modular bucket-type structure of conceptual hydrologic models with machine learning. Plainly spoken, Neural ODE models are conceptual hydrologic models with deep learning cores. The presented models M50 and M100 depict hydrologic implementations of the general Neural ODE approach (Chen et al., 2018; Rackauckas et al., 2020) - and up to our knowledge the first ones in hydrology. The substitution of constitutive functions by neural networks has shown to significantly increase predictive performance compared to a plain conceptual model while keeping the same natural physical constraints. Overall, hydrologic Neural ODE models perform similarly well to or better than state-of-the-art pure or partial machine-learning models, but overcome three different limitations of former approaches as introduced in Section 1:

First, using the conceptual hydrologic model structure preserves the interpretability of the model as traditionally given by conceptual models and appreciated by the hydrologic community. Internal model states and processes can directly be inspected for plausibility, and their physical interpretation fosters system understanding. The Neural ODE approach might further trigger advancement in a more fundamental manner of building "conceptual" models: Theoretically, modellers only need to set up the conceptual framework but do not have to specify parametrizations within the model and let the neural networks learn plausible relations. Potentially, even features that are often neglected in typical conceptual models, like hysteresis (Gharari and Razavi, 2018), could be elicited.

Second, the Neural ODE allows for continuous time solutions. In principle, this also allows to include data on an irregular temporal resolution for both training and testing. Physical principles and mechanistic structure act as guide rails that are naturally included and do not have to be learned or enforced as with pure machine learning approaches. At the same time, the method is flexible enough to learn constitutive relations from data.

Third, our approach invites prior physical knowledge to be incorporated into the model. For instance, the Neural ODE approach allows to include processes that are fully known as hard-coded features like a sewage treatment plant discharging into the stream at a known temporal pattern. Locally, expert knowledge might be available about hydrologic systems that can be accounted for. Pure data-driven methods might not be able to infer this knowledge from data alone and pure mechanistic models



might not provide the desired flexibility like Neural ODE models.

In principle, the introduced approach can be applied to any conceptual hydrologic model. Numerous alternative bucket-type
models and frameworks exist that can be fused with neural networks partially or entirely. The number of states and processes
is adjustable according to specific requirements of the modelling problem at hand, or in a more generic setup for multiple

catchments. Already the used EXP-Hydro model as rather simplistic example of conceptual model facilitated a drastic im-
provement of model performance when used as basis for Neural ODE models. Many sophisticated conceptual models exist
(like SAC-SMA) that could serve as a framework for also more sophisticated hydrologic Neural ODE models.

With the hydrological Neural ODE model we seek to introduce a tool in-between existing top-down and bottom-up ap-

proaches that paves the way for various subsequent research routes. For example, the deterministic model can be made proba-
bilistic to enable uncertainty assessments as currently performed for stochastic hydrologic models (Reichert et al., 2021). Also,
due to its generic setup, the Neural ODE approach appears to be suitable for being trained with multiple basins simultaneously
including static attributes like in respective investigations for LSTM models (Kratzert et al., 2019c; Feng et al., 2020; Jiang
et al., 2020). This large-sample hydrology setting might be particularly useful to further investigate process relations data-

scarce variable ranges. We will investigate this in a subsequent step.

*Code availability.*  All software was written in the programming language Julia. A working example will be made available in the near future.

**Appendix A:**

**A1   EXP-Hydro Equations**

The simple rainfall-runoff model EXP-Hydro (Patil and Stieglitz, 2014) comprises of only two state variables representing
buckets, five mechanistic processes and 6 parameters $\Theta$ (see Table A1). There are three inputs to the model: length of day
$L_{day}$, temperature $T$ and precipitation $P$.





**Table A1.** EXP-hydro parameter definitions, meaning and units (cf. Patil and Stieglitz, 2014)

| Parameter | Original definition | Meaning | Units |
|:---:|:---:|:---|:---:|
| $\Theta_1$ | $T_{min}$ | snow fall temperature | $^\circ C$ |
| $\Theta_2$ | $T_{max}$ | snow melt temperature | $^\circ C$ |
| $\Theta_3$ | $D_f$ | thermal degree-day factor | $mm(day \cdot ^\circ C)^{-1}$ |
| $\Theta_4$ | $S_{max}$ | maximum water storage | $mm$ |
| $\Theta_5$ | $f$ | runoff decline rate | $mm^{-1}$ |
| $\Theta_6$ | $Q_{max}$ | maximum subsurface runoff | $mm \cdot day^{-1}$ |

For ease of readability and comparability to Jiang et al. (2020), parameters are written like in Table A1 as they were originally defined by Patil and Stieglitz (2014). Further, the storage state $S_{snow}$ is written as $S_0$ and $S_{water}$ is written as $S_1$. For ease of readability dependence on time is implicitly assumed and $t$ is dropped. Moreover, the driving forces and model states relevant for each process are explicitly named. Hence, the processes in EXP-Hydro are formulated as:

- Precipitation as snow or rain:

$$P_{snow}(P,T;T_{min}) = \begin{cases} 0 & T > T_{min} \\ P & \text{otherwise} \end{cases} \tag{A1}$$

$$P_{rain}(P,T;T_{min}) = \begin{cases} P & T > T_{min} \\ 0 & \text{otherwise} \end{cases} \tag{A2}$$

- Evapotranspiration:

$$ET(T,L_{day},S_1;S_{max}) = \begin{cases} PET(T,L_{day}) \cdot (S_1/S_{max}) & 0 \le S_1 \le S_{max} \\ PET(T,L_{day}) & S_1 > S_{max} \end{cases} \tag{A3}$$

originally using Hamon's formula (Hamon, 1963) for potential evapotranspiration $PET(T,L_{day}) = 29.8 \cdot L_{day} \frac{e_{sat}(T)}{T+273.2}$, with saturation water pressure $e_{sat}(T) = 0.611 \cdot exp\left(\frac{17.3T}{T+237.3}\right)$. Note that $L_{day}$ is factored out in model M0 for $\widetilde{ET}$. There, we use $\widetilde{PET}(T) = PET(T,L_{day})/L_{day}$.

- Melting:





$$M(T, S_0; T_{max}, D_f) = \begin{cases} min(S_0, D_f \cdot (T - T_{max})) & T > T_{max} \text{ and } S_0 > 0 \\ 0 & \text{otherwise} \end{cases} \tag{A4}$$

– Discharge:

$$Q(S_1; f, Q_{max}, S_{max}) = Q_{bucket}(S_1, f, Q_{max}, S_{max}) + Q_{spill}(S_1, S_{max}) \tag{A5}$$

with

$$Q_{bucket}(S_1; f, Q_{max}, S_{max}) = \begin{cases} Q_{max} \cdot exp(-f \cdot (S_{max} - S_1)) & 0 \leq S_1 \leq S_{max} \\ Q_{max} & S_1 > S_{max} \end{cases} \tag{A6}$$

and

$$Q_{spill}(S_1; S_{max}) = \begin{cases} 0 & 0 \leq S_1 \leq S_{max} \\ S_1 - S_{max} & S_1 > S_{max} \end{cases} \tag{A7}$$

## A2 Rationales behind M50 and M100

With the substitutions from M0 to M50, we want to highlight two important features of the Neural ODE modelling approach. First, physical knowledge can directly be included into the model: The ET prescription uses potential evapotranspiration based on Hamon's formula (Hamon, 1963), in which length of day $L_{day}$ is factored (see Appendix A1). This is a fully accessible input variable to the model - for a certain latitude and time, it is a physically fixed information (that theoretically could also be calculated within the model). When used in a multiplication as in the chosen ET prescription, it can therefore simply be kept as a factor and only the rest of the ET formula has to be substituted and learned by a neural network. It is a plausible assumption, that ET is proportional to the length of day as represented in the mechanistic description of Hamon's formula referring to the light-activation of plants' stomata. This proportionality is therefore kept in M50 when substituting the rest of the ET formula by a neural network. As can be seen in the model scheme in Figure 1 (a), the $NN_{ET}$ does not obtain $L_{day}$ as input but instead $S_{snow}$ as there could be interference of snow cover with evapotranspiration.

Second, in hydrologic models, discharge is often split up into (at least) a base flow component and an excess or peak flow component that acts above a certain threshold of the water storage. In the Neural ODE approach, these two flow components



can be substituted by a neural network with a single output node because neural networks are particularly suited to learn non-linearities. Hence, rather than defining an "artificial" threshold beyond which a new process is added, NNs can learn a continuous relation between water storage and model inputs to discharge. Unlike the Q formula in M0, we added precipitation

as second input to the $NN_Q$ in M50 to potentially account for direct runoff.

M50 is meant to demonstrate how strongly predictive performance can be increased by including some more flexible, data-driven model parts, i.e. only partial modifications within the traditional modelling approach. This approach is similar to the one in Bennett and Nijssen (2021) although there fixed time-stepping was applied, only one internal process was substituted and

the exact same inputs were given to the NN as were given to the mechanistic process. Further, their goal was not to ultimately enhance stream flow prediction but to substitute the internal process, i.e. the turbulent heat flux.

In the next step from M50 to M100, the other mechanistic processes that are "hard-coded" in the plain EXP-Hydro are also substituted. These are to distinguish between precipitation as rain or snow and the melting process that transfers water from

the snow storage unit to the main storage unit. As opposed to $ET$ and $Q$, over certain parts of the year, these processes are not occurring, e.g. if all snow was molten in spring, there is no melting process going on in summer. Hence the Neural ODE model has to learn these regime differences. Again, $L_{day}$ is factored out in the ET process, which highlights a feature of the Neural ODE approach: If $L_{day}$ shall be included it could also be given to the NN as input. Yet, the NN could also learn a relation between $L_{day}$ and $Q$ which is physically implausible. In a plain machine-learning approach, this specific use of $L_{day}$ cannot

be as easily assigned.

## A3   SAC-SMA

The current benchmark hydrologic model for the CAMELS-US dataset is the Sacramento-soil moisture accounting model (SAC-SMA; see  Newman et al., 2015,  and references therein). The simulated discharge values from the SAC-SMA model

used for evaluation are taken from the CAMELS dataset (Addor et al., 2017) (discharge predictions for test period 1.10.2000 - 30.9.2010).

Note, however, that training and testing periods for the SAC-SMA were different from those used here. The SAC-SMA was calibrated with a split-sample approach where 30 years of data (1.10.1980 to 30.9.2010) were split up into two parts each

covering 15 years. For details refer to Newman et al. (2015). In contrast, we used the first 20 years for training and the last 10 years for testing. The scores of NSE, FHV and mNSE for the SAC-SMA model shown in Figure A1 are evaluated for this 10 years testing period. Hence, the results should only be considered as indication and not as strict assessment when being directly compared to the results in Section 3.





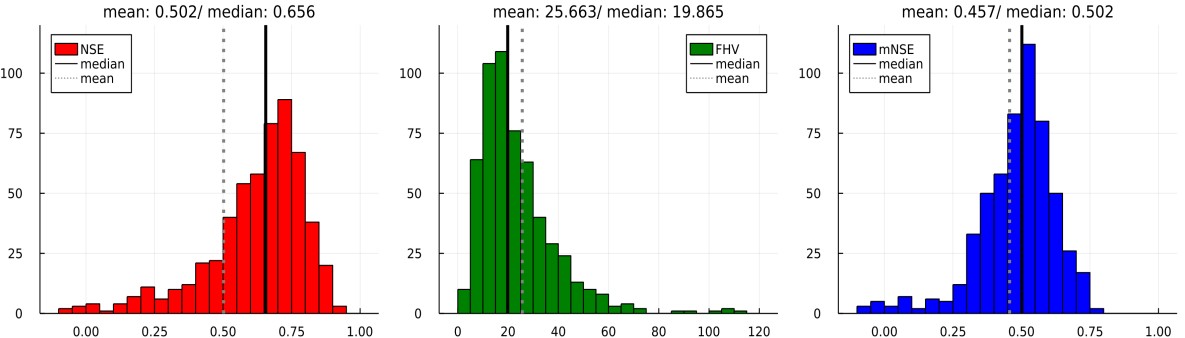

**Figure A1.** Histograms of NSE (red; optimal value: 1), FHV (green; optimal value: 0) and mNSE (blue; optimal value: 1) for the SAC-SMA model over 569 basins.

Figure shows the overall performance of the SAC-SMA model on all the 569 basins for the testing period. It is significantly better than the simple conceptual EXP-Hydro model implemented as M0 and it achieves comparable levels of performance compared to the partial and pure machine-learning models evaluated (see Section 3). Yet, it does not score better than these although many more processes and interrelations (and corresponding assumptions) were put into the model: 20 of in total 35 parameters were calibrated and the rest adjusted according to expert knowledge (Newman et al., 2015). This demonstrates that, in principle, conceptual models do have the ability to reach high scores in model rating - but come with comparably high effort in setup, tuning and adjustment compared to pure or hybrid machine learning based methods.

*Author contributions.*  M.H. had the original idea and developed the conceptualization and methodology of the study. M.H. developed the software with initial support by A.S.. M.H. conducted all model simulations and their formal analysis. Results were discussed and further research steps planned between C.A., M.BJ., A.S., F.F. and M.H.. The visualizations and the original draft of the manuscript were prepared by M.H., reviewing and editing was provided by M.BJ., C.A., A.S. and F.F.. Funding was acquired by F.F.. All authors have read and agreed to the current version of the manuscript.

*Competing interests.*  The authors declare that there is no conflict of interest.

*Acknowledgements.*  The authors would like to thank Shijie Jiang for providing the original results from Jiang et al. (2020) for the model benchmarking in this work. We thank Peter Reichert for fruitful discussions and suggestions that helped improving the manuscript.




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
