# Peer review of "Improving hydrologic models for predictions and process understanding using Neural ODEs"

_Hydrology and Earth System Sciences, 2022_

## Referee Comment (RC2)

165

[referee-annotated manuscript omitted]

---

## Author Response (AR1)

**Author replies (HESS-2022-56)**

**21. August 2022**

*We thank the two referees and participant of the community discussion for the detailed reviews, constructive feedback and valuable comments.*

*Please find our answers below in italic. In most cases, our answers are the same as the replies we gave in the interactive discussion. We conducted changes during the revision as we suggested them and as they were confirmed by the AE. Those parts of our replies below that changed from our original replies are highlighted in bold.*

**1. Replies to the referees:**

**Referee 1:**

**General Comments**

This study uses Neural ODEs to address the gap between process-based models (PBMs) and artificial neural networks/deep learning, a hot topic in current research efforts in hydrology. While the former allows physical interpretation and fosters process/domain knowledge, the latter often shows higher simulation accuracy but lacks the aforementioned interpretability. This is an interesting and well performed study where the authors show the potential of Neural ODEs to combine both domains while simultaneously improving the simulation accuracy. The manuscript is well written and well structured.  I like especially the structure of the introduction which is divided into three parts, presenting the state of the art of the major domains (PBMs and Neural Networks) and how they can be brought together by Neural ODEs. The study in general addresses a relevant scientific question within the scope of HESS.

- *We thank the referee for the thorough review, detailed feedback and valuable comments. Please find our replies corresponding to the enumerated points.*

Some weaknesses (in my opinion) are addressed in the following.

**Specific Comments**

1. While I like the general structure of the introduction, it sometimes lacks the bigger picture. While it is not necessary to elaborate on the bigger picture at the very beginning (as on modeling in hydrology in general, for example), the whole chapter will in my opinion profit from shedding light on other approaches that aim to combine PBMs and DL approaches such as: PGDL (physically guided deep learning) /PIML (physically informed machine learning) and PINNs. Especially the latter might be worth a look, because they also target differential equations. This should help to really justify the claim in L90: "Yet, none of the pure or novel hybrid machine learning approaches has addressed all the above gaps regarding interpretability, physics and knowledge at once."

- ***Thank you for the suggestion. We added a new paragraph to section "1.3 Scientific Machine Learning and Neural ODEs" that introduces the bigger picture and takes up the suggested approaches.***

2. I disagree with the statement that LSTMs generally work well on daily timescales. In my opinion this depends on the modeling task, the data, the system etc. So, while it might be true that peak flows are not captured by LSTM models in some basins, I think with higher temporal resolution this might change to a certain extent. Also, the work by Gauch et al. 2021 is rather an approach to use different data sources than to overcome some model shortcoming. The authors should justify their claim by citing examples or by elaborating why that is in their opinion.

- *Thank you for raising these points in the discussion. We assume that the referee refers to l. 40ff in the manuscript. We agree that modelling success is always subject to task, data, system, etc. Yet, for predictive tasks at daily resolution, LSTMs have frequently proven to show overall best results. **We have added more specifics to the respective paragraph.** We do think that Gauch et al. 2021 ("Rainfall–Runoff Prediction at Multiple Timescales with a Single Long Short-Term Memory Network") is the appropriate reference on temporal resolution of LSTM predictions since their work introduces Multi-Timescale LSTMs for this purpose. We think the issue on using different data sources is addressed in Kratzert, F., Klotz, D., Hochreiter, S., and Nearing, G. S.: A note on leveraging synergy in multiple meteorological data sets with deep learning for rainfall–runoff modeling, Hydrology and Earth System Sciences, 25, 2685–2703, 2021. that we have also in our list of references. If we misunderstand the referees comment here, we appreciate further remarks in this respect.*

3. I am wondering why $NN^{50}\_Q$ did not receive T as an additional input. That would be an obvious choice for a better estimation of ET. Maybe the authors could better elaborate why the chose additional inputs for some of the NNs (such as P to infer runoff) and not for others.

In the M50 approach, two small, process-specific NNs are used to substitute two different processes: one for discharge Q and the other one for evapotranspiration ET. Qualitatively, temperature T should not have a direct impact on Q but indirectly on the entire water balance via ET (and of course melting M but M is not substituted by an NN in M50). We considered only precipitation P and water storage $S\_water$ to have an effect on Q and therefore these are the only input variables to $NN^{50}\_Q$ – this is how we show it in Figure 1 (b). Yet, the referee's remark made us aware of the fact that in the top of Figure 1 (b), P is depicted as input variable to the other neural network $NN^{50}\_ET$. This should rather be temperature T as it is also used in the mechanistic formulation of ET in Equation A3 and as it is discussed in the manuscript. We thank the referee for spotting typo and **we changed the figure accordingly. Please also see our reply to the next comment**

4. I think figure 1 is good in general. However, the authors might think of combining the scheme of subplot a with clear indications for (b) and (c), where, e.g., to better explain, which paths/arrows are replaced by which NN model.

- **Thank you. We replaced the original figure by the requested more detailed figure of which we think readers might benefit more.**

5. L190 ff. (just as an example): I am missing some details on how the training exactly works.

First, did you take measures to prevent overfitting while pre-training the NN models? What did you do exactly, if not, please explain why.

→ *We did not take measures to prevent overfitting during pre-training. We did not see a need for any sort of regularization in the pre-training phase since this step is only supposed to "roughly" inform the NN parameters such that the NN learns how to process the input variables appropriately to approximate the internal model processes. The target values in this*

*step stem from hard-coded processes relations from the original conceptual and are themselves only approximations to what actually happens in reality. We minimized the sum of squared residuals to these target values as objective function over 1000 iterations in this pre-training phase.* **We added more explanation to the corresponding text in section "2.3 Procedure and Model Rating".**

Second, after reading I am still quite unsure how the Neural ODE is trained as a whole. Are the NNs fine-tuned in this step? If yes, how? What is, for example, the Loss function, is it a multi-step procedure which separates the NN training from the overall calibration or is it somehow combined? I appreciated the appendix in general, however, in this regard it did not help me to answer my questions.

- *We are happy to provide more details w.r.t. the training: The loss function for the training of the whole Neural ODE models (conceptual model frame + NN(s) within) is NSE. The training is conducted using the GalacticOptim.jl package in Julia where the Neural ODE model parameters (i.e., the parameters of the NN(s) within the conceptual model frame) are optimized using ADAM(): For M100 the only model parameters in the model are the weights and biases of the $NN^{100}$. For M50, the model parameters that are optimized are the weights and biases of the $NN^{50}\_Q$ and $NN^{50}\_ET$ (the model parameters in the process $M(t)$ are those from the conceptual model and are kept fixed.) Apart from this optimization, there is no additional fine-tuning.* **We added more explanation to the corresponding text in section "2.3 Procedure and Model Rating".**

   *Overall we think that the release of the code will help also with this point (please see our answer to point 12)*

6. I generally like the stringent way of comparing with the results from Jiang et al. 2020.

- *Thank you. We also found Jiang et al., 2020 to serve as a good reference point for a comparison.*

7. The authors should improve the readability of figure 3. Where data points are plotted, I hardly recognize the rest of the plot. Maybe another color-concept might help in general, also to avoid dotted and dashed lines, which are hard to read in my opinion (because the gap between dashs and dots is almost larger than some of the peaks). I generally recommend to select different colors that separate well from each other, and use slight transparency to ensure readability in overlapping regions. In my opinion, being aware of colorblind readers is more useful than ensuring readability when printed/or when black&white. Maybe check colorbrewer or related sites for ideas.

- Thank you, this is a very important point. We agree that all readers should be able to easily read the shown figures. Since the major purpose of these figures was to show the ability of the models to match the temporal pattern of hydrographs, snow water equivalent and (model-based) water storage rather than the exact fitting of data-points, we decided that gaps in dashed and dotted lines do not hinder this insight. **We reiterated the figures and improved it according to the referee's suggestions.**

8. You might also check if the colormaps in figures 5 to 7 could be improved in terms of readability for colorblind people ( I suspect this is already okay – not sure)

- Thank you, this very important point raised by the referee relates to comment 7 above. We chose the color-coding "thermal" having in mind that it is an often recommended set of

colors. Yet, now we read "The misuse of color in science communication" by Crameri et al., 2020, Nature Communications, to obtain better knowledge in this field. There, "thermal" is highlighted as one of the colorblindness-friendly options confirming our choice.

9. L287: are they that similar? Probably, but due to the scale it is hard to recognize. Did you explore also log-scale plots? Did this provide additional insights in this regard?

- **Thank you for your suggestion. We ended up adding the log-scale plots since they allow us to provide a more extensive discussion and to highlight insights in more detail in section "4 Internal States and Processes of Neural ODE models".**

10. Figures 5 to 7: As you discuss yourself, what you do is using neural networks for strong extrapolation. Of course this is an application, where ANNs fail on a regular basis. So while I think that this is not necessarily the case, we simply do not know. I would recommend to better discuss this aspect. In L. 395, you state that "20 years of data is not enough to extrapolate towards these limits". I would counter that even with more data this is not necessarily possible. Please discuss the general problems with using ANNs/DL for extrapolation and why you might think that this applies (or not) for your application and analysis.

- Thank you for the discussion of this aspect. We agree that extrapolation of neural networks is a challenging topic. In our application case we purposely also picked basins to show that this extrapolation is difficult and sometimes leads to counter-intuitive behavior – to highlight these issues. We still think that more data might help because, first, they help the network to refine relations for data ranges that are closer to the ranges of pure extrapolation. Second, since we consider natural systems where physical limits constrain the problem to be learned by the NN. These are enforced by the chosen model structure and act as regularization aside of the information provided by data (please see our answer w.r.t. regularization for remark 5 above). We think that this combination might help to elicit extrapolation relations that can then be evaluated in plausibility testing. An example for this would be a very high rainfall intensity where our data might never cover a centennial occurrence, but we would still be able to qualitatively judge whether the relation learned by network is plausible or not. Nonetheless, we agree, more data is not necessarily the solution to the problem alone. An extended discussion of this topic is useful and we will elaborate on it. **We changed the discussion in section "5.2 Internal States and Processes" accordingly.**

11. Please revise your wording when speaking of significance. Lines: 317, 328, 345, 358, 360 … (and others). If you did perform a statistical test please add the information, otherwise, use e.g., "considerably".

- We partially agree. There can be confusion if "significance" is used with respect to results from statistical tests, but we do not think that this is the case here. Significant is a frequently used term also outside of only statistical tests and this was our intention to use it. Otherwise, we would have used "statistically significant". **Nonetheless, we checked the different instances and substituted most occurrences.**

12. The software code should be made available. This might have helped to answer some of my questions and will foster the application of this approach in the future.

- *We made a Github repository available.*

**Technical Corrections**

1. Statement in L. 204-205: "With NSE < 0, the model is worse than just using the model 205 average, […]" – NSE compares to the observed average, not to the model average.

- Thank you for spotting this. As we correctly showed in Equation 3, the text should of course say "observed average" – **we corrected this.**

2. I find it quite unintuitive to mix NSE, mNSE and CoE_alpha. NSE is more common in hydrology, why not only name it consistently like this?

- We agree that directly showing and using the equation for NSE is very common in hydrology. Yet, we purposely decided to focus on CoE_alpha and to show that NSE (like the also used mNSE) is only a special case of CoE_alpha. This is not often highlighted in hydrology but we think it is an useful relation to know. We think that lines 203-213 explain this sufficiently and hope that the referee can follow our intention here.

3. L331 spelling: evapotranspiration

- Thank you, **we corrected the typo.**

**Referee 2:**

I found the this paper very interesting. The present approach is novel in terms of using Neural ODEs for streamflow modeling. My recommendation for this paper is a "minor revision". My comments are given in the attached pdf file. My future recommendation for the authors of this paper is to adopt the Neural ODEs in watershed experiencing different climatic conditions across the world.

*We thank the referee for constructive remarks and we appreciate the future recommendation to advance our research. Providing a modelling approach that allows to extrapolate to different climatic conditions is one of the motivations to use NeuralODEs. We have collected all comments from the pdf file that was attached and we have answered them below:*

l.3 Replace the highlighted with "not yet fully understood"

- **We agree, we changed "given" to "understood".**

l.9 Please mention the exact number (609?)
- **We replaced "several hundred" by the exact number: 569**

l.12 There is no need to divide the "Introduction" section through sub-sections. Hence you may remove 1.1, 1.2 sub sections.
- We introduction brings together several big topics that are important for the remainder of the article. Our intention to use sub-sections was to help structuring this. Referee 1 commented to like this setup. We see this as confirmation of our assumption that this structure helps to keep a red line throughout section 1. Therefore, our preference is to keep the sub-sections and we hope that the referee can follow our argument here.

l.16 (PUB)
- We think the referee refers to separating the abbreviation PUB from the references. We seek to avoid two parenthesis following one another. **We replaced "," by ";" for better distinction.**

l.18 You can start this sentence with "In this paper"
- Thank you for the suggestion. Since we did not refer to "this paper" anywhere else and in order to keep this style of writing**, we started the sentence with "Here,…"**

l.21 It would be better if you can support this sentence through literature
**- We added supporting literature.**

l.22 «Although, the»
- We think there is no comma necessary here.

l.22-23 "Especially, when»
- Thanks for highlighting this. After reading this sentence again**, we dropped "Especially" entirely, since it appears only to be a filling word here.**

l.31 I suggest the authors to brief some of the internals whicj Nearing et al., & ... have explained (May be in 1 or 2 sentences)
- **Thank you, we added some details here**, e.g. the referenced papers demonstrate linking hidden states in LSTM models to physical variables like soil moisture content.

l.41-46 Thank you for looking into the different words that are made from word combinations. We looked into the individual cases and learned the following:

Timescales
- following https://www.collinsdictionary.com/dictionary/english/timescale : "timescale" and "time scale" both exist, but refer to different contexts. In our case, timescales seems to be appropriate
Continuous-time
- Agreed, we will change it to "continuous time". Also, following this, we will delete the hyphen in "hidden-states"
Step-wise
- Agreed, we will change it to "stepwise"
Real-world
- following https://grammarhow.com/real-world-or-real-world/ : we keep real-world with a hyphen because it is used as adjective to a noun
Time-continuous
- we will replace this by "continuous in time"

**We made the suggested changes.**

l.61 Provide some examples of certain bucket-type models? (HBV-light?)
- **we added examples like the mentioned HBV or GR4J**

l.74 Spell out the acronymn «FUSE»
- **Agreed, we added "Framework for Understanding Structural Errors"**

l.129 Material and Methods
- We see that "Material and Methods" is often the default header for this section. Since we do not specify any material as it is used, e.g., in a lab, we prefer keeping "Methods" only.

l.168 Spell out the abbreviation "CAMELS"
- **Agreed, we added "Catchment Attributes and Meteorology for Large-sample Studies"**

l.188 Was there a specific reason to use NSE. What is the advantage of using NSE over Kling Gupta Efficiency?
- We purposely only used the metrics that were also employed in our reference study by Jiang et al. (2020), i.e. NSE, FHV and mNSE. Since NSE is a commonly used training and testing metric in hydrology, we did not see a need to switch to KGE. Yet, we know that KGE with its three different parts is generally a good option for hydrologic model rating.

L.215 This has been mistakenly written here ("FHV")
- It shall be in the parenthesis, but at the beginning, not the end. This is a TeX-based error, thank you for spotting it! **We changed it.**

l.258-259Give a basic description about these two catchments.
- **We added a short description providing some context information about the two catchments to the beginning of section "4 Internal States and Processes of Neural ODE models".**

2. **Replies in the community discussion**

This is an interesting article. There are minor issues that I think would increase the value of it (in no particular order):

- M(t) (the flow between the two buckets in M0, which I guess is a snow melting flow) is not formally defined in the model description.

*We introduce melting M in l.134. and in l.148, and we reference appendix A1 where M is defined for the original conceptual model in Equation A4. We are open for further specification w.r.t. what the formal definition might be lacking but we are sure that it exists.*

- There is no explanation of why different errors where applied to different modules of the hybrid models. Ther eis a comment in the discussions regarding the known failures of models to capture discharge peaks, this performance is strongly linked to the type of error (or the noise distribution in a sotchastic modeling context) that was used to train the model. Did the chosen erros for traing improve this? why? what was the criteria to choose the different errors?

*There was no error function assigned. The models are deterministic and where trained using the available data points. We applied the same procedure as was done in the reference study by Jiang et al., 2020, in order to ensure comparability.* ***We added further explanation to the training steps in section "2.3 Procedure and Model Rating".***

- I could not find a link to the software, and "be makde available in the near future" is too ambiguous. The software should be part of the publication work, citing:

  "An article about computational result is advertising, not scholarship. The actual scholarship is the full software environment, code and data, that produced the result." -- Buckheit and Donoho

***We made a Github repository available.***

- The data-driven relation learn by M50/100 are clearly tuned to the data. Assuming the proposed mechanism is causal and universal, wouldn't it then make more sense to train these modules in the totality of the data, not per catchment? On the one hand, it is well stablished that non-causal data-driven can easily ourperform causal models (e.g. a casual structure X -> Y -> Z with noise in X larger than in Z will cause data-driven models to choose Z to as the best predcitor of Y, alas non-causal). On the other hand, it is unlikely that NN models will use relations that go beyond the scope of the data, hence the optinal relations found per catchment might be reflecting circumstantial relations, but the mechanisms proposed are suposed to be principled mechanisms, not cimcustantial.

*We agree. This is where our current research is headed. We work on training the models on multiple basins. One of the points we want to investigate is exactly whether the learned relations go beyond the scope of the data. The physical structure of the model enforces a sort of regularization that we think might help in this respect.* ***We modified the third paragraph in section "6 Conclusions and Outlook".***